# Learning Recurrent Span Representations for Extractive Question Answering

**Kenton Lee**[*]
University of Washington
Seattle, WA
kentonl@cs.washington.edu

**Tom Kwiatkowski   Ankur Parikh   Dipanjan Das**
Google
New York, NY
{tomkwiat, aparikh, dipanjand}@google.com

## Abstract

The reading comprehension task, that asks questions about a given evidence document, is a central problem in natural language understanding. Recent formulations of this task have typically focused on *answer selection* from a set of candidates pre-defined manually or through the use of an external NLP pipeline. However, Rajpurkar et al. (2016) recently released the SQUAD dataset in which the answers can be arbitrary strings from the supplied text. In this paper, we focus on this *answer extraction* task, presenting a novel model architecture that efficiently builds fixed length representations of all spans in the evidence document with a recurrent network. We show that scoring explicit span representations significantly improves performance over other approaches that factor the prediction into separate predictions about words or start and end markers. Our approach improves upon the best published results of Wang & Jiang (2016) by 5% and decreases the error of Rajpurkar et al.'s baseline by $> 50\%$.

## 1 Introduction

A primary goal of natural language processing is to develop systems that can answer questions about the contents of documents. The reading comprehension task is of practical interest – we want computers to be able to read the world's text and then answer our questions – and, since we believe it requires deep language understanding, it has also become a flagship task in NLP research.

A number of reading comprehension datasets have been developed that focus on answer selection from a small set of alternatives defined by annotators (Richardson et al., 2013) or existing NLP pipelines that cannot be trained end-to-end (Hill et al., 2016; Hermann et al., 2015). Subsequently, the models proposed for this task have tended to make use of the limited set of candidates, basing their predictions on mention-level attention weights (Hermann et al., 2015), or centering classifiers (Chen et al., 2016), or network memories (Hill et al., 2016) on candidate locations.

Recently, Rajpurkar et al. (2016) released the less restricted SQUAD dataset[1] that does not place any constraints on the set of allowed answers, other than that they should be drawn from the evidence document. Rajpurkar et al. proposed a baseline system that chooses answers from the constituents identified by an existing syntactic parser. This allows them to prune the $O(N^2)$ answer candidates in each document of length $N$, but it also effectively renders $20.7\%$ of all questions unanswerable.

Subsequent work by Wang & Jiang (2016) significantly improve upon this baseline by using an end-to-end neural network architecture to identify answer spans by labeling either individual words, or the start and end of the answer span. Both of these methods do not make independence assumptions about substructures, but they are susceptible to search errors due to greedy training and decoding.

In contrast, here we argue that it is beneficial to simplify the decoding procedure by enumerating all possible answer spans. By explicitly representing each answer span, our model can be globally normalized during training and decoded exactly during evaluation. A naive approach to building the $O(N^2)$ spans of up to length $N$ would require a network that is cubic in size with respect

---

[*]Work completed during internship at Google, New York.
[1]http://stanford-qa.com

to the passage length, and such a network would be untrainable. To overcome this, we present a novel neural architecture called RASOR that builds fixed-length span representations, *reusing* recurrent computations for shared substructures. We demonstrate that directly classifying each of the competing spans, and training with global normalization over all possible spans, leads to a significant increase in performance. In our experiments, we show an increase in performance over Wang & Jiang (2016) of $5\%$ in terms of exact match to a reference answer, and $3.6\%$ in terms of predicted answer F1 with respect to the reference. On both of these metrics, we close the gap between Rajpurkar et al.'s baseline and the human-performance upper-bound by $> 50\%$.

## 2 EXTRACTIVE QUESTION ANSWERING

### 2.1 TASK DEFINITION

Extractive question answering systems take as input a question $\mathbf{q} = \{q_0, \ldots, q_n\}$ and a passage of text $\mathbf{p} = \{p_0, \ldots, p_m\}$ from which they predict a single answer span $\mathbf{a} = \langle a_{start}, a_{end} \rangle$, represented as a pair of indices into $\mathbf{p}$. Machine learned extractive question answering systems, such as the one presented here, learn a predictor function $f(\mathbf{q}, \mathbf{p}) \rightarrow \mathbf{a}$ from a training dataset of $\langle \mathbf{q}, \mathbf{p}, \mathbf{a} \rangle$ triples.

### 2.2 RELATED WORK

For the SQUAD dataset, the original paper from Rajpurkar et al. (2016) implemented a linear model with sparse features based on $n$-grams and part-of-speech tags present in the question and the candidate answer. Other than lexical features, they also used syntactic information in the form of dependency paths to extract more general features. They set a strong baseline for following work and also presented an in depth analysis, showing that lexical and syntactic features contribute most strongly to their model's performance. Subsequent work by Wang & Jiang (2016) use an end-to-end neural network method that uses a Match-LSTM to model the question and the passage, and uses pointer networks (Vinyals et al., 2015) to extract the answer span from the passage. This model resorts to greedy decoding and falls short in terms of performance compared to our model (see Section 5 for more detail). While we only compare to published baselines, there are other unpublished competitive systems on the SQUAD leaderboard, as listed in footnote 5.

A task that is closely related to extractive question answering is the Cloze task (Taylor, 1953), in which the goal is to predict a concealed span from a declarative sentence given a passage of supporting text. Recently, Hermann et al. (2015) presented a Cloze dataset in which the task is to predict the correct entity in an incomplete sentence given an abstractive summary of a news article. Hermann et al. also present various neural architectures to solve the problem. Although this dataset is large and varied in domain, recent analysis by Chen et al. (2016) shows that simple models can achieve close to the human upper bound. As noted by the authors of the SQUAD paper, the annotated answers in the SQUAD dataset are often spans that include non-entities and can be longer phrases, unlike the Cloze datasets, thus making the task more challenging.

Another, more traditional line of work has focused on extractive question answering on sentences, where the task is to extract a sentence from a document, given a question. Relevant datasets include datasets from the annual TREC evaluations (Voorhees & Tice, 2000) and WikiQA (Yang et al., 2015), where the latter dataset specifically focused on Wikipedia passages. There has been a line of interesting recent publications using neural architectures, focused on this variety of extractive question answering (Tymoshenko et al., 2016; Wang et al., 2016, *inter alia*). These methods model the question and a candidate answer sentence, but do not focus on possible candidate answer *spans* that may contain the answer to the given question. In this work, we focus on the more challenging problem of extracting the precise answer span.

## 3 MODEL

We propose a model architecture called RASOR[2] illustrated in Figure 1, that explicitly computes embedding representations for candidate answer spans. In most structured prediction problems (e.g. sequence labeling or parsing), the number of possible output structures is exponential in the input

---

[2]An abbreviation for Recurrent Span Representations, pronounced as *razor*.

length, and computing representations for every candidate is prohibitively expensive. However, we exploit the simplicity of our task, where we can trivially and tractably enumerate all candidates. This facilitates an expressive model that computes joint representations of every answer span, that can be globally normalized during learning.

In order to compute these span representations, we must aggregate information from the passage and the question for every answer candidate. For the example in Figure 1, RASOR computes an embedding for the candidate answer spans: *fixed to*, *fixed to the*, *to the*, etc. A naive approach for these aggregations would require a network that is cubic in size with respect to the passage length. Instead, our model reduces this to a quadratic size by reusing recurrent computations for shared substructures (i.e. common passage words) from different spans.

Since the choice of answer span depends on the original question, we must incorporate this information into the computation of the span representation. We model this by augmenting the passage word embeddings with additional embedding representations of the question.

In this section, we motivate and describe the architecture for RASOR in a top-down manner.

## 3.1 SCORING ANSWER SPANS

The goal of our extractive question answering system is to predict the single best answer span among all candidates from the passage $\mathbf{p}$, denoted as $\mathbf{A}(\mathbf{p})$. Therefore, we define a probability distribution over all possible answer spans given the question $\mathbf{q}$ and passage $\mathbf{p}$, and the predictor function finds the answer span with the maximum likelihood:

$$f(\mathbf{q}, \mathbf{p}) := \underset{\mathbf{a} \in \mathbf{A}(\mathbf{p})}{\operatorname{argmax}} P(\mathbf{a} \mid \mathbf{q}, \mathbf{p}) \qquad (1)$$

One might be tempted to introduce independence assumptions that would enable cheaper decoding. For example, this distribution can be modeled as (1) a product of conditionally independent distributions (binary) for every word or (2) a product of conditionally independent distributions (over words) for the start and end indices of the answer span. However, we show in Section 5.2 that such independence assumptions hurt the accuracy of the model, and instead we only assume a fixed-length representation $h_{\mathbf{a}}$ of each candidate span that is scored and normalized with a softmax layer (**Span score** and **Softmax** in Figure 1):

$$s_{\mathbf{a}} = w_a \cdot \text{FFNN}(h_{\mathbf{a}}) \qquad\qquad \mathbf{a} \in \mathbf{A}(\mathbf{p}) \qquad (2)$$

$$P(\mathbf{a} \mid \mathbf{q}, \mathbf{p}) = \frac{\exp(s_{\mathbf{a}})}{\sum_{\mathbf{a}' \in \mathbf{A}(\mathbf{p})} \exp(s_{\mathbf{a}'})} \qquad\qquad \mathbf{a} \in \mathbf{A}(\mathbf{p}) \qquad (3)$$

where FFNN$(\cdot)$ denotes a fully connected feed-forward neural network that provides a non-linear mapping of its input embedding, and $w_a$ denotes a learned vector for scoring the last layer of the feed-forward neural network.

## 3.2 RASOR: RECURRENT SPAN REPRESENTATION

The previously defined probability distribution depends on the answer span representations, $h_{\mathbf{a}}$. When computing $h_{\mathbf{a}}$, we assume access to representations of individual passage words that have been augmented with a representation of the question. We denote these question-focused passage word embeddings as $\{p_1^*, \ldots, p_m^*\}$ and describe their creation in Section 3.3. In order to reuse computation for shared substructures, we use a bidirectional LSTM (Hochreiter & Schmidhuber, 1997) to encode the left and right context of every $p_i^*$ (**Passage-level BiLSTM** in Figure 1). This allows us to simply concatenate the bidirectional LSTM (BiLSTM) outputs at the endpoints of a span to jointly encode its inside and outside information (**Span embedding** in Figure 1):

$$\{p_1^{*\prime}, \ldots, p_m^{*\prime}\} = \text{BILSTM}(\{p_1^*, \ldots, p_m^*\}) \qquad (4)$$

$$h_{\mathbf{a}} = [p_{a_{start}}^{*\prime}, p_{a_{end}}^{*\prime}] \qquad\qquad \langle a_{start}, a_{end} \rangle \in \mathbf{A}(\mathbf{p}) \qquad (5)$$

where BILSTM$(\cdot)$ denotes a BiLSTM over its input embedding sequence and $p_i^{*\prime}$ is the concatenation of forward and backward outputs at time-step $i$. While the visualization in Figure 1 shows a single layer BiLSTM for simplicity, we use a multi-layer BiLSTM in our experiments. The concatenated output of each layer is used as input for the subsequent layer, allowing the upper layers to depend on the entire passage.

### 3.3 QUESTION-FOCUSED PASSAGE WORD EMBEDDING

Computing the question-focused passage word embeddings $\{p_1^*, \ldots, p_m^*\}$ requires integrating question information into the passage. The architecture for this integration is flexible and likely depends on the nature of the dataset. For the SQUAD dataset, we find that both passage-aligned and passage-independent question representations are effective at incorporating this contextual information, and experiments will show that their benefits are complementary. To incorporate these question representations, we simply concatenate them with the passage word embeddings (**Question-focused passage word embedding** in Figure 1).

We use fixed pretrained embeddings to represent question and passage words. Therefore, in the following discussion, notation for the words are interchangeable with their embedding representations.

**Question-independent passage word embedding**  The first component simply looks up the pretrained word embedding for the passage word, $p_i$.

**Passage-aligned question representation**  In this dataset, the question-passage pairs often contain large lexical overlap or similarity near the correct answer span. To encourage the model to exploit these similarities, we include a fixed-length representation of the question based on soft alignments with the passage word. The alignments are computed via neural attention (Bahdanau et al., 2014), and we use the variant proposed by Parikh et al. (2016), where attention scores are dot products between non-linear mappings of word embeddings.

$$s_{ij} = \text{FFNN}(p_i) \cdot \text{FFNN}(q_j) \qquad 1 \le j \le n \qquad (6)$$

$$a_{ij} = \frac{\exp(s_{ij})}{\sum_{k=1}^{n} \exp(s_{ik})} \qquad 1 \le j \le n \qquad (7)$$

$$q_i^{align} = \sum_{j=1}^{n} a_{ij} q_j \qquad (8)$$

**Passage-independent question representation**  We also include a representation of the question that does not depend on the passage and is shared for all passage words.

Similar to the previous question representation, an attention score is computed via a dot-product, except the question word is compared to a universal learned embedding rather any particular passage word. Additionally, we incorporate contextual information with a BiLSTM before aggregating the outputs using this attention mechanism.

The goal is to generate a coarse-grained summary of the question that depends on word order. Formally, the passage-independent question representation $q^{indep}$ is computed as follows:

$$\{q_1', \ldots, q_n'\} = \text{BILSTM}(\mathbf{q}) \qquad (9)$$

$$s_j = w_q \cdot \text{FFNN}(q_j') \qquad 1 \le j \le n \qquad (10)$$

$$a_j = \frac{\exp(s_j)}{\sum_{k=1}^{n} \exp(s_k)} \qquad 1 \le j \le n \qquad (11)$$

$$q^{indep} = \sum_{j=1}^{n} a_j q_j' \qquad (12)$$

where $w_q$ denotes a learned vector for scoring the last layer of the feed-forward neural network.

This representation is a bidirectional generalization of the question representation recently proposed by Li et al. (2016) for a different question-answering task.

Given the above three components, the complete question-focused passage word embedding for $p_i$ is their concatenation: $p_i^* = [p_i, q_i^{align}, q^{indep}]$.

### 3.4 LEARNING

Given the above model specification, learning is straightforward. We simply maximize the log-likelihood of the correct answer candidates and backpropagate the errors end-to-end.

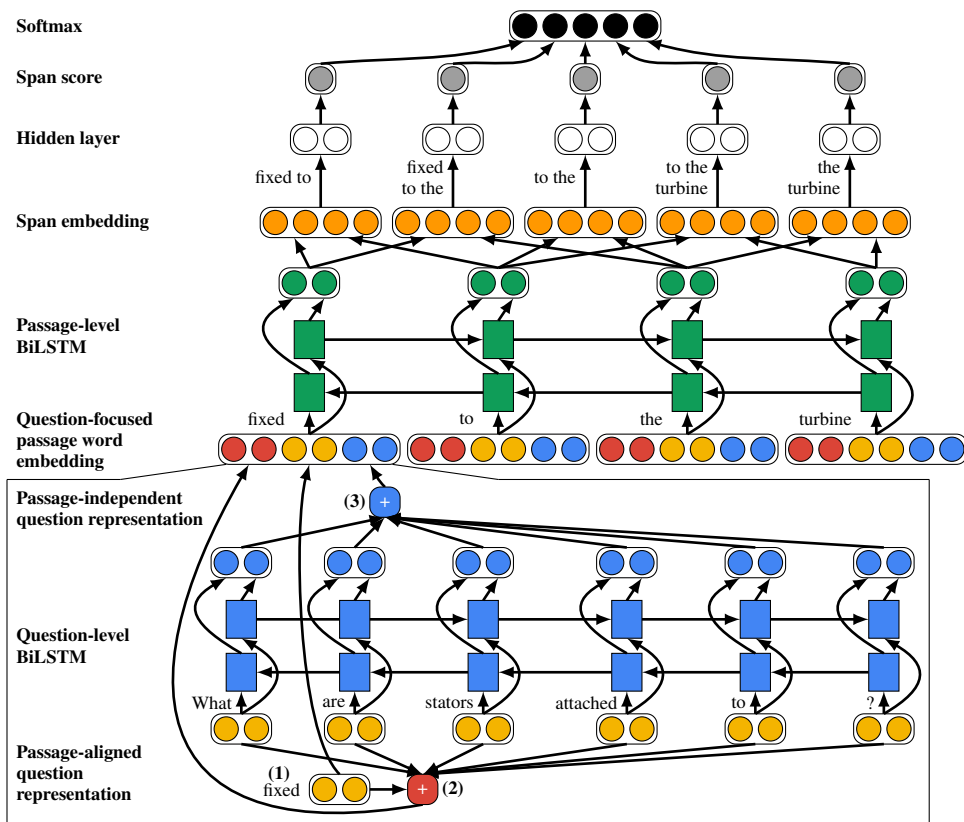

Figure 1: A visualization of RASOR, where the question is *"What are the stators attached to?"* and the passage is *"...fixed to the turbine ..."*. The model constructs question-focused passage word embeddings by concatenating **(1)** the original passage word embedding, **(2)** a passage-aligned representation of the question, and **(3)** a passage-independent representation of the question shared across all passage words. We use a BiLSTM over these concatenated embeddings to efficiently recover embedding representations of all possible spans, which are then scored by the final layer of the model.

## 4 EXPERIMENTAL SETUP

We represent each of the words in the question and document using 300 dimensional GloVe embeddings trained on a corpus of $840$bn words (Pennington et al., 2014). These embeddings cover $200$k words and all out of vocabulary (OOV) words are projected onto one of $1$m randomly initialized $300$d embeddings. We couple the input and forget gates in our LSTMs, as described in Greff et al. (2016), and we use a single dropout mask to apply dropout across all LSTM time-steps as proposed by Gal & Ghahramani (2016). Hidden layers in the feed-forward neural networks use rectified linear units (Nair & Hinton, 2010). Answer candidates are limited to spans with at most 30 words.

To choose the final model configuration, we ran grid searches over: the dimensionality of the LSTM hidden states $(25, 50, 100, 200)$; the number of stacked LSTM layers $(1, 2, 3)$; the width $(50, 100, 150, 200)$ and depth $(1, 2)$ of the feed-forward neural networks; the dropout rate $(0, 0.1, 0.2)$; and the decay multiplier $(0.9, 0.95, 1.0)$ with which we multiply the learning rate every $10$k steps. The best model uses a single $150$d hidden layer in all feed-forward neural networks; $50$d LSTM states; two-layer BiLSTMs for the span encoder and the passage-independent question representation; dropout of $0.1$ throughout; and a learning rate decay of $5\%$ every $10$k steps.

All models are implemented using TensorFlow[3] and trained on the SQuAD training set using the ADAM (Kingma & Ba, 2015) optimizer with a mini-batch size of $4$ and trained using 10 asynchronous training threads on a single machine.

---

[3] www.tensorflow.org

## 5 RESULTS

We train on the 80k (question, passage, answer span) triples in the SQuAD training set and report results on the 10k examples in the SQuAD development set. Due to copyright restrictions, we are currently not able to upload our models to Codalab[4], which is required to run on the blind SQuAD test set, but we are working with Rajpurkar et al. to remedy this, and this paper will be updated with test numbers as soon as possible.

All results are calculated using the official SQuAD evaluation script, which reports exact answer match and F1 overlap of the unigrams between the predicted answer and the closest labeled answer from the 3 reference answers given in the SQuAD development set.

### 5.1 COMPARISONS TO OTHER WORK

Our model with recurrent span representations (RASOR) is compared to all previously published systems [5]. Rajpurkar et al. (2016) published a logistic regression baseline as well as human performance on the SQuAD task. The logistic regression baseline uses the output of an existing syntactic parser both as a constraint on the set of allowed answer spans, and as a method of creating sparse features for an answer-centric scoring model. Despite not having access to any external representation of linguistic structure, RASOR achieves an error reduction of more than 50% over this baseline, both in terms of exact match and F1, relative to the human performance upper bound.

|  | Dev | | Test | |
| --- | --- | --- | --- | --- |
| System | EM | F1 | EM | F1 |
| Logistic regression baseline | 39.8 | 51.0 | 40.4 | 51.0 |
| Match-LSTM (Sequence) | 54.5 | 67.7 | 54.8 | 68.0 |
| Match-LSTM (Boundary) | 60.5 | 70.7 | 59.4 | 70.0 |
| RASOR | 66.4 | 74.9 | – | – |
| RASOR (Ensemble) | 68.2 | 76.7 | – | – |
| Human | 81.4 | 91.0 | 82.3 | 91.2 |

Table 1: Exact match (EM) and span F1 on SQuAD. We are currently unable to evaluate on the blind SQuAD test set due to copyright restrictions. We confirm that we did not overfit the development set via 5-fold cross validation of the hyper-parameters, resulting in $66.0 \pm 1.0$ exact match and $74.5 \pm 0.9$ F1.

More closely related to RASOR is the *boundary model* with Match-LSTMs and Pointer Networks by Wang & Jiang (2016). Their model similarly uses recurrent networks to learn embeddings of each passage word in the context of the question, and it can also capture interactions between endpoints, since the end index probability distribution is conditioned on the start index. However, both training and evaluation are greedy, making their system susceptible to search errors when decoding. In contrast, RASOR can efficiently and explicitly model the quadratic number of possible answers, which leads to a 14% error reduction over the best performing Match-LSTM model.

We also ensemble RASOR with a baseline model described in Section 5.2 that independently predicts endpoints rather than spans (Endpoints prediction in Table 2b). By simply computing the product of the output probabilities, this ensemble further increases performance to 68.2% exact-match. We examine the causes of this improvement in Section 6.

Since we do not have access to the test set, we also present 5-fold cross validation experiments to demonstrate that our dev-set results are not an outcome of overfitting through hyper-parameter selection. In this 5-fold setting, we create 5 pseudo dev/test splits from the SQuAD development set.[6] We choose hyper-parameters on the basis of the pseudo dev set, and report performance on the disjoint pseudo test set. Each of the pseudo dev sets led us to choose the same optimal model

---

[4]http://stanford-qa.com

[5]As of submission, other unpublished systems are shown on the SQuAD leaderboard, including *Match-LSTM with Ans-Ptr (Boundary+Ensemble)*, *Co-attention*, *r-net*, *Match-LSTM with Bi-Ans-Ptr (Boundary)*, *Co-attention old*, *Dynamic Chunk Reader*, *Dynamic Chunk Ranker with Convolution layer*, *Attentive Chunker*.

[6]We split by Wikipedia page ID and use $\frac{4}{5}$ as a development set and $\frac{1}{5}$ as a test set.

hyper-parameters from a grid of 59 settings, as well as very similar training stopping points. We compute the mean and standard deviation of both evaluation metrics for these optimal models on the pseudo test set, resulting in a $66.0 \pm 1.0$ exact match and $74.5 \pm 0.9$ F1. These results show that our hyper-parameter selection procedure is not overfitting on the 10k SQUAD development set, and we subsequently expect that our model's performance will translate to the SQUAD test set.

## 5.2 MODEL VARIATIONS

We investigate two main questions in the following ablations and comparisons. (1) How important are the two methods of representing the question described in Section 3.3? (2) What is the impact of learning a loss function that accurately reflects the span prediction task?

**Question representations**  Table 2a shows the performance of RASOR when either of the two question representations described in Section 3.3 is removed. The passage-aligned question representation is crucial, since lexically similar regions of the passage provide strong signal for relevant answer spans. If the question is only integrated through the inclusion of a passage-independent representation, performance drops drastically. The passage-independent question representation over the BiLSTM is less important, but it still accounts for over $3\%$ exact match and F1. The input of both of these components is analyzed qualitatively in Section 6.

| Question representation | EM | F1 |
|---|---|---|
| Only passage-independent | 48.7 | 56.6 |
| Only passage-aligned | 63.1 | 71.3 |
| RASOR | 66.4 | 74.9 |

(a) Ablation of question representations.

| Learning objective | EM | F1 |
|---|---|---|
| Membership prediction | 57.9 | 69.7 |
| BIO sequence prediction | 63.9 | 73.0 |
| Endpoints prediction | 65.3 | 75.1 |
| Span prediction w/ log loss | 65.2 | 73.6 |

(b) Comparisons for different learning objectives given the same passage-level BiLSTM.

Table 2: Results for variations of the model architecture presented in Section 3.

**Learning objectives**  Given a fixed architecture that is capable of encoding the input question-passage pairs, there are many ways of setting up a learning objective to encourage the model to predict the correct span. In Table 2b, we provide comparisons of some alternatives (learned end-to-end) given only the passage-level BiLSTM from RASOR. In order to provide clean comparisons, we restrict the alternatives to objectives that are trained and evaluated with exact decoding.

The simplest alternative is to consider this task as binary classification for every word (*Membership prediction* in Table 2b). In this baseline, we optimize the logistic loss for binary labels indicating whether passage words belong to the correct answer span. At prediction time, a valid span can be recovered in linear time by finding the maximum contiguous sum of scores.

Li et al. (2016) proposed a sequence-labeling scheme that is similar to the above baseline (*BIO sequence prediction* in Table 2b). We follow their proposed model and learn a conditional random field (CRF) layer after the passage-level BiLSTM to model transitions between the different labels. At prediction time, a valid span can be recovered in linear time using Viterbi decoding, with hard transition constraints to enforce a single contiguous output.

We also consider a model that independently predicts the two endpoints of the answer span (*Endpoints prediction* in Table 2b). This model uses the softmax loss over passage words during learning. When decoding, we only need to enforce the constraint that the start index is no greater than the end index. Without the interactions between the endpoints, this can be computed in linear time. Note that this model has the same expressivity as RASOR if the span-level FFNN were removed.

Lastly, we compare with a model using the same architecture as RASOR but is trained with a binary logistic loss rather than a softmax loss over spans (*Span prediction w/ logistic loss* in Table 2b).

The trend in Table 2b shows that the model is better at leveraging the supervision as the learning objective more accurately reflects the fundamental task at hand: determining the best answer span.

First, we observe general improvements when using labels that closely align with the task. For example, the labels for *membership prediction* simply happens to provide single contiguous spans in the supervision. The model must consider far more possible answers than it needs to (the power set of all words). The same problem holds for *BIO sequence prediction*– the model must do additional work to learn the semantics of the BIO tags. On the other hand, in RASOR, the semantics of an answer span is naturally encoded by the set of labels.

Second, we observe the importance of allowing interactions between the endpoints using the span-level FFNN. RASOR outperforms the *endpoint prediction* model by 1.1 in exact match, The interaction between endpoints enables RASOR to enforce consistency across its two substructures. While this does not provide improvements for predicting the correct *region* of the answer (captured by the F1 metric, which drops by 0.2), it is more likely to predict a clean answer span that matches human judgment exactly (captured by the exact-match metric).

## 6 ANALYSIS

Figure 2 shows how the performances of RASOR and the endpoint predictor introduced in Section 5.2 degrade as the lengths of their predictions increase. The endpoint predictor underpredicts single word answer spans, while overpredicting answer spans with more than 8 words.

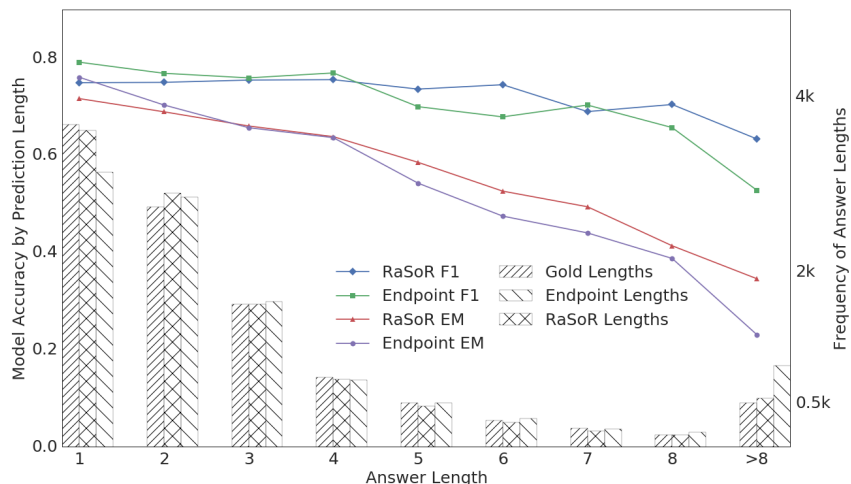

Figure 2: F1 and Exact Match accuracy of RASOR and the endpoint predictor over different predictions lengths, along with the distribution of both models' prediction lengths and the gold answer lengths.

Since the endpoints predictor does not explicitly model the interaction between the start and end of any given answer span, it is susceptible to choosing the span start and end points from separate answer candidates. For example, consider the following endpoints prediction that is most different in length from a correct span classification. Here, the span classifier correctly answers the question 'Where did the Meuse flow before the flood?' with 'North Sea' but the endpoints prediction is:

> 'south of today's line Merwede-Oude Maas to the North Sea and formed an archipelago-like estuary with Waal and Lek. This system of numerous bays, estuary-like extended rivers, many islands and constant changes of the coastline, is hard to imagine today. From 1421 to 1904, the Meuse and Waal merged further upstream at Gorinchem to form Merwede. For flood protection reasons, the Meuse was separated from the Waal through a lock and diverted into a new outlet called "Bergse Maas", then Amer and then flows into the former bay Hollands Diep'

In this prediction, we can see that both the start 'south of . . . ' and the end '. . . Hollands Diep' have a reasonable answer type. However, the endpoints predictor has failed to model the fact that they cannot resonably be part of the same answer, a common error case. The endpoints predictor predicts 514 answers with $> 25$ more words than the gold answer, but the span classifier never does this.

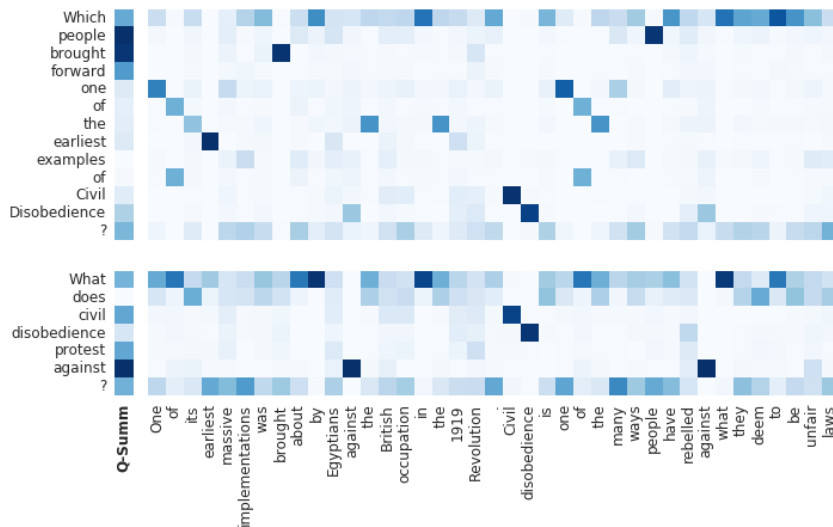

Figure 3: Attention masks from RASOR. Top predictions are 'Egyptians', 'Egyptians against the British', and 'British' in the first example and 'unjust laws', 'what they deem to be unjust laws', and 'laws' in the second.

| Most attended | Question |
|---|---|
| **conditions** | In Gebhard v Consiglio...Milano, the requirements to be registered in Milan before being able to practice law would be allowed under what **conditions**? |
| **Were** | **Were** the restored tapes able to have color added to them to enhance the picture or did they remain black and white? |
| **Did** | **Did** the European Court of Justice rule the defendant in the case of Commission v. Edith Cresson broke any laws? |
| **whom** | The church holds that they are equally bound to respect the sacredness of the life and well-being of **whom**? |
| **Whose** | **Whose** thesis states that the solution to a problem is solvable with reasonable resources assuming it allows for a polynomial time algorithm ? |
| **language** | What **language** did the Court of Justice accept to be required to teach in a Dublin college in Groner v Minister for Education? |
| **term** | Income not from the creation of wealth but by grabbing a larger share of it is know to economists by what **term**? |

Table 3: Example questions and their most attended words in the passage-independent question representation (Equation 11). These examples have the greatest attention (normalized by the question length) in the development set. The attention mechanism typically seeks words in the question that indicate the answer type.

Figure 3 shows attention masks for both of RASOR's question representations. The passage-independent question representation pays most attention to the words that could attach to the answer in the passage ('brought', 'against') or describe the answer category ('people'). Meanwhile, the passage-aligned question representation pays attention to similar words. The top predictions for both examples are all valid syntactic constituents, and they all have the correct semantic category. However, RASOR assigns almost as much probability mass to it's incorrect third prediction 'British' as it does to the top scoring correct prediction 'Egyptian'. This showcases a common failure case for RASOR, where it can find an answer of the correct type close to a phrase that overlaps with the question – but it cannot accurately represent the semantic dependency on that phrase.

A significant architectural difference from other neural models for the SQUAD dataset, such as Wang & Jiang (2016), is the use of the question-independent passage representation (Equation 12). Table 3 shows examples in the development set where the model paid the most attention to a single word in the question. The attention mechanism tends to seek words in the question that indicate the answer type, e.g. '**language**' from the question: 'What **language** did the Court of Justic accept ...' This pattern provides insight for the necessity of using *both* question representations, since the answer type information is orthogonal to passage alignment information.

## 7 CONCLUSION

We have shown a novel approach for perform extractive question answering on the SQUAD dataset by explicitly representing and scoring answer span candidates. The core of our model relies on a recurrent network that enables shared computation for the shared substructure across span candidates. We explore different methods of encoding the passage and question, showing the benefits of including both passage-independent and passage-aligned question representations. While we show that this encoding method is beneficial for the task, this is orthogonal to the core contribution of efficiently computing span representation. In future work, we plan to explore alternate architectures that provide input to the recurrent span representations.

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
