# Peer review of "Learning Recurrent Span Representations for Extractive Question Answering"

_ICLR 2017 — rejected_

[Official Review · AnonReviewer1 · rating 6 · confidence 5 · 16 Dec 2016]

This paper presents an architecture for answer extraction task and evaluates on the SQUAD dataset. The proposed model builds fixed length representations of all spans in the answer document based on recurrent neural network. It outperforms a few baselines in exact match and F1 on SQUAD.

It is unfortunate that the blind test results are not obtained yet due to the copyright issue. There are quite a few other systems/submissions on the SQUAD leader board that were available for comparison.

Given that there's no result on the test set reported, the grid search for hyperparameters on the dev set directly is also a concern, even though the authors did cross validation experiments.

[Official Review · AnonReviewer2 · rating 6 · confidence 3 · 16 Dec 2016]

The authors proposed RASOR to address the problem of finding the best answer span according to a given question. The focus of the paper is mainly on how to model the relationship between question and the answer spans. The idea proposed by this paper is reasonable, but not ground breaking. The analysis is interesting and potentially useful. I would hope the authors can go extra miles to analyze different choices of boundary prediction models and make a more convincing case for the necessity of modeling the score of the span globally.

The main idea behind RASOR is to globally normalize and rank the scores of the possible answer spans. RASOR is able to achieve this by first modeling the hidden vectors of all words with LSTMs. Then, the representation of a text span is formed by concatenating the corresponding hidden vectors of the start and the end word of the corresponding chunk. The approach is reasonable, but not earth shattering. Also, the table 6 shows that the improvement over end-prediction point is not very large.

I appreciate the fact that authors conduct several analysis experiments as some of them are quite interesting. For example, it seems that question independent representation is also very import to the performance. In addition to the current analysis, I also want to get a clear idea on what makes the current model be better than the Match-LSTM. Is it hyper-parameter tuning? Or it is due to the use of the question independent representation?

Another good thing about the proposed model is that it is relatively simple, so there is a chance that the proposed techniques can be combined with other newly proposed ones.

[Official Review · AnonReviewer3 · rating 7 · confidence 4 · 17 Dec 2016]

This paper proposes RaSoR, a method to efficiently representing and scoring all possible spans in an extractive QA task. While the test set results on SQuAD have not been released, it looks likely that they are not going to be state-of-the-art; with that said, the idea of enumerating all possible spans proposed in this paper could potentially improve many architectures. The paper is very well-written and the analysis/ablations in the final sections are mostly interesting (especially Figure 2, which confirms what we would intuitively believe). Based on its potential to positively impact other researchers working on SQuAD, I recommend that the paper is accepted.

[Author Response · Kenton Lee · 13 Jan 2017]
**Review response**

We thank all three reviewers for the valuable comments and suggestions. 

We agree with reviewer 1 that the lack of test results is not ideal and sadly we do not yet have a manner in which we can run on the hidden test set, as not all of our code is open-sourced. However, we hope that the significant dev set size of 10k items, along with the cross-validation results add some reassurance that our hyperparameter tuning scheme has not overfit the data.

Reviewer 3 points out that the results in this paper are no longer state of the art. It is true that there are other papers on the leaderboard that have now surpassed our results, largely through ensembling. However, we believe that our paper is the only work to specifically study the impact of different span representations and we agree with Reviewer 3 that our findings should be complementary to other recent work on this dataset. We have added some extra quantitative and qualitative analysis of the differences between the span classifier and the endpoints predictor to illustrate the manner in which the quality of endpoint predictions degrade for longer sentences, in particular showing the tendency of endpoint models to pick out endpoints from separate answer candidates.

Reviewer 2 points out that the difference in performance between our model and the Match-LSTM cannot be accounted for by the difference in label type alone, and asks for the other most salient differences between the two approaches. While there are many small differences between the two implementations, the ablations in Table 2.a. suggest that most of this gap is accounted for by the passage independent question representation that is missing in the Match-LSTM. We have added an analysis of this representation in a new Table 3 and we have updated our discussion of the Match LSTM to clarify the basis of our comparison.

[Final Decision · Program Chairs · 06 Feb 2017]
**ICLR committee final decision**

The program committee appreciates the authors' response to concerns raised in the reviews. Unfortunately, most reviewers are not leaning sufficiently towards acceptance. In particular, it is unfortunate that authors can not evaluate their model on the leaderboard due to copyright issues. The role of standard datasets and benchmarks is to allow for meaningful comparisons. Evaluation on non-standard splits defeats this purpose. Fortunately, sounds like authors are working on getting their model evaluated on the leaderboard. Resolving that and incorporating reviewers' feedback will help make the paper stronger.